# Striatal Isolated from *Cyathus striatus* Extracts Induces Apoptosis in Human Pancreatic Cancer Cells

**DOI:** 10.3390/molecules27092746

**Published:** 2022-04-24

**Authors:** Fuad Fares, Soliman Khatib, Jacob Vaya, Lital Sharvit, Einav Eizenberg, Solomon Wasser

**Affiliations:** 1Department of Human Biology, Faculty of Natural Sciences, University of Haifa, Haifa 3498838, Israel; lsharvit@univ.haifa.ac.il (L.S.); einave.seidner@gmail.com (E.E.); 2Laboratory of Natural Compounds and Analytical Chemistry, MIGAL Galilee Research Institute, Kiryat Shmona 11016, Israel; solimankh@migal.org.il (S.K.); vaya@migal.org.il (J.V.); 3Tel-Hai College, Tel-Hai 1220800, Israel; 4Institute of Evolution and Department of Evolutionary and Environmental Biology, Faculty of Natural Sciences, University of Haifa, Haifa 3498838, Israel; spwasser@research.haifa.ac.il

**Keywords:** medicinal mushroom, *Cyathus striatus*, striatal, pancreatic cancer, apoptosis

## Abstract

The aim of the present study was to identify the structure of active compounds in *Cyathus stratus* that previously demonstrated anti-pancreatic cancer activity. The active compounds were purified from a crude extract by a series of RP-18 preparative chromatography using homemade octadecyl silica gel column. HPLC injection of the crude extract revealed a chromatogram with three main peaks with retention times (RT) 15.6, 18.2, and 22.5 min. Each fraction that exhibited promising activity in vitro was further separated using various available chromatographic techniques. The purified compound with the ultimate anti-cancer activity appeared at RT of 15.8 in the HPLC chromatogram with more than 90% purity. The main peak at the mass spectra appeared at *m*/*z* = 446.2304 with the calculated molecular formula of C_25_H_34_O_7_. One- and two-dimensional NMR analyses indicated that the structure of the active molecule (peak 15.8 min in HPLC) was identified as striatal C. Exposure of human pancreatic cancer cells to purified striatal C resulted in induction of apoptosis. Further studies are needed in order to develop a method for the synthesis of striatal in order to use it in clinical studies for treatment of cancer.

## 1. Introduction

Natural products have been the most important source of drugs in history [1]. Many edible mushrooms were used in traditional folk medicine while others are used only for their medicinal properties [2,3,4,5]. Medicinal mushrooms were used as natural compounds to modulate immune responses and possess anti-cancer, anti-microbial, and anti-oxidant properties [6,7]. In addition, it was reported that oyster mushrooms, commonly known as *Pleurotus ostreatus*, induced apoptosisin human colon cancer cells (HCT-116) in a dose-dependent manner (12.5 to 200 μg/mL) [8]. Moreover, it was reported for the first time that the oyster mushroom *Pleurotus florida* mediated trimetallic nanoparticle synthesis via a rapid, simple, environmentally friendly, economical, and feasible method [9]. Mushrooms are described as a rich source of bioactive compounds and contain primary and secondary metabolites, where the secondary metabolites exhibit medicinal properties [10].

Many pharmaceutical substances with potent and unique properties have been extracted from mushrooms and anti-cancer remedies were prepared from these extracts, such as the following polysaccharides: lentinan, krestin, and schizophyllan [11]. Higher Basidiomycetes mushrooms (HBMs) represent a major and still largely unexploited source of new pharmaceutical products. It was suggested that HBMs act as immunomodulators or biological response modifiers (BRMs). This advantage is one of the reasons why they are currently used as adjuncts to cancer treatment in many countries [12]. Fungal metabolites have been gaining scientific interest for their medicinal properties. Studies on medicinal mushrooms proved their potential not only as dietary supplements and immunomodulators [13,14], but also as modulators of various cellular responses [15]. Low-molecular-weight (LMW) substances with fungal origins can penetrate cell membranes and interfere, in particular, with cellular pathways linked to processes such as inflammation, carcinogenesis, cell differentiation survival, and metastasis. Characteristic changes such as deregulation of cell cycle machinery, self-sufficiency in growth signals, insensitivity to growth inhibitory signals, evasion of apoptosis, tissue invasion, metastasis, and sustained angiogenesis are hallmarks of majority of cancers and can be found in pancreatic ductal adenocarcinoma (PDAC) [16,17,18,19].

*Cyathus striatus* (*CS*) is a higher Basidiomycetes mushroom of the family Nidulariaceae. It has widespread distribution throughout temperate regions of the world. *CS* is a rich source of bioactive chemical compounds. It was reported that *CS* produces indole substances, complexes of diterpenoid antibiotic compounds, and cyathins [20]. The indole substances, which were later named striatins (A, B, and C), have antibiotic activity against imperfect fungi and various Gram-positive and Gram-negative bacteria [21]. However, most bioactive substances isolated from mushrooms are high molecular weights, (HMWs) such as polysaccharides. Our interest is in LMW compounds that are capable of exhibiting anti-tumor activity. Previously, we found that ethyl acetate extracts from *CS* media induced apoptosis in human pancreatic cancer cells through extrinsic and intrinsic apoptotic pathways [22,23]. In addition, gene expression analysis revealed the involvement of mitogen-activated protein kinase (MAPK) and P53 signaling pathways and pointed toward endoplasmic reticulum stress-induced apoptosis. Moreover, *CS* extracts significantly inhibited tumour growth in vivo using animal models [23]. Thus, we have focused on the search for novel LMW compounds in *CS* extracts that induce apoptosis in human pancreatic cancer cells, and therefore, might be useful in the search of new therapies for pancreatic cancer.

## 2. Materials and Methods

### 2.1. Extract Production

The mycelium of the *CS* mushroom was grown first on solid medium and then it was transferred onto submerged conditions, as described before [24]. The specific strain was grown in submerged conditions for biomass production for a period of 10 days, long enough to reach the growth stage of secondary metabolite production. The growing media of the mushroom, culture liquid (CL), was extracted with ethyl acetate (EAC) in ratio of 1000 mL (CL):500 mL (EAC) [25]. The dried extract was diluted with 99.9% dimethyl sulfoxide (DMSO) (Sigma-Aldrich, St. Louis, MO, USA) to reach a concentration of 50 mg/mL and it was kept at −20 °C until use.

### 2.2. Isolation and Structure Elucidation of the Bioactive Compound

The active compounds were purified from the extract (0.5 g of crude extract) by a series of RP-18 preparative chromatography using homemade reverse-phase octadecyl silica gel column. The solvents used to eluate the extract fractions were (A)—double distilled water (DDW) and (B)—acetonitrile (ACN). The chromatography started with 0% of solvent B, then it increased by 5% every 10 min to 100% B, and it was kept at 100% B for another 30 min. Flow rate was 5 mL/min to obtain 5 mg of the most active compound which appeared at 16 min in the HPLC chromatogram.

### 2.3. HPLC Analysis

The chromatographed fractions were analyzed by injecting 20 µL of the fraction solution dissolved in methanol (2 mg mL^−1^) into a UHPLC connected to a photodiode array detector (Dionex Ultimate 3000) with a reverse-phase column (Phenomenex RP-18, 150 4.0 mm, 3 μm). The mobile phase consisted of (A) DDW with 0.1% formic acid and (B) acetonitrile containing 0.1% formic acid, run at a gradient starting from 40% B, then increased to 90% B for 22 min, and kept at 90% B for another 8 min, at a flow rate of 1 mL min^−1^.

### 2.4. LC–MS Analysis

LC–MS analysis was performed with an electrospray ionization (Dual AJS ESI+) source connected to a UHD accurate-mass Q-TOF LC–MS 6540A (Agilent Technologies). The ESI capillary voltage was set to 4000 V, the fragmenter to 150 V, gas temperature to 300 °C, gas flow to 8 mL/min, and nebulizer to 35 psi. The mass spectra (*m*/*z* 100–1000) were acquired in positive-ion mode.

### 2.5. NMR Analysis

The isolated compound was dissolved in deuterated chloroform (CD_3_Cl) and NMR spectra were recorded at room temperature with AV-III 600 Bruker spectrometer. The ^13^C-NMR spectra were recorded using zgig and dept-135; 2D-NMR methods included COSY, HSQC, and HMBC. 

### 2.6. Cell Culture

The human pancreatic cancer cell lines—HPAF-II, well differentiated, and PL45, poorly differentiated (ATCC, Rockville, MD, USA)—were maintained in MEM-EAGLE and DMEM media, respectively, supplemented with 1% L-glutamine, 10% fetal calf serum (FCS), and 1% PenStrep (penicillin + streptomycin) (Biological industries, Kibbutz Beit-Haemek, Israel). The HPAF-II cell line was supplemented with an additional 1% sodium pyruvate, and the PL45 cell line was supplemented with an additional 1% L-glutamine. Cells were grown in a humidified incubator at 37 °C with 5% CO_2_ in air and were fed twice a week with fresh media. 

### 2.7. Cell Proliferation

Evaluation of the fungal extract effect on cell line viability was performed using an XTT assay (Biological Industries, Kibbutz Beit-Haemek, Israel). HPAF-II and PL45 cells (10^4^) were seeded in 100 µL of medium using 96-well plates. After 24 h, the fungal extract was added (1, 2.5, 5, 7.5, 10, 15, and 50 µg/mL) for 4, 8, 12, 24, 48, and 72 h, respectively. After this, viability level was determined according to the manufacturer’s instructions using an Elisa reader (BioTek) at 450 nm and subtracted from the reference absorbance at 620 nm. Survival was compared to that of vehicle-treated control cells. The experiments were repeated 2–5 times independently and conducted in at least 3 replicates. Data was presented as the average proliferation percentage of the respective control.

### 2.8. Effect of CS Extract on Lactate Dehydrogenase (LDH) Release

LDH is a cytoplasmic enzyme that catalyzes the oxidation of L-lactate to pyruvate with NAD^+^ as a hydrogen acceptor—the final step in the metabolic chain of anaerobic glycolysis. The extracellular appearance of LDH serves as a marker for tissue lysis since cell damage, such as necrosis, causes a rise in LDH in the cells’ medium [21,26]. In order to exclude the possibility for cytotoxic effects of the extract on the cell lines, LDH leakage into the medium was measured in aliquots of the extracellular fluid of each sample using an LDH Cytotoxicity Detection Kit (Roche Diagnostics, Mannheim, Germany). 

### 2.9. Cell Cycle Analysis

A total of 10^6^ HPAF-II and PL45 cells were treated with selected doses of the *CS* extract or with the purified fractions for a treatment period that was determined according to the XTT results. Then, the cells were trypsinized and collected with the growth medium, centrifuged, washed with PBS, and fixed with 70% ethanol for one hour. This was followed by incubation with 0.1% NP-40 for 5 min in 4 °C followed by incubation on ice with 100 µg/mL RNase for 30 min. Finally, 50 µg/mL PI was added for 20 min. Cell cycle phase distribution was determined by Fluorescence Activated Cell Sorter (FACS) flow cytometry (Becton Dickinson, Franklin Lakes, NJ, USA); 10,000 cells were counted for each control and treatment group.

### 2.10. Apoptosis

Apoptotic cells were detected by several methods as follows.

#### 2.10.1. Annexin-V

Cell apoptosis was measured using Annexin V-FITC Apoptosis Detection Kit (MBL, Woburn, MA, USA). Briefly, 10^6^ cells were seeded in T-25 flasks. On the next day, HPAF-II and PL45 cells were treated with selected doses of the *CS* extract or fractions for 4 h. At the end of treatment time, cells were trypsinized and collected with the growth medium, centrifuged, washed with PBS, and 2 × 10^5^ cells were counted and collected for the analysis. The cells were resuspended in 85 μL of 1X binding buffer; 10 μL of Annexin V-FITC and 5 μL of propidium iodide were added. Cells were incubated for 15 min at room temperature in the dark prior to analysis. Apoptosis cells were determined using a BD Facscalibor flow cytometer (Becton Dickinson, USA).

#### 2.10.2. TUNEL Assay

Cell morphology was examined using TUNEL (In Situ Cell Death Detection Kit, Roch) and DAPI staining. The cells were washed twice with PBS and fixed for 60 min. Subsequently, the cells were incubated with a TUNEL reaction mixture that contained TdT and fluorescein-dUTP. During this incubation period, TdT catalyzes the addition of fluorescein-dUTP at the free 3′-OH groups in single- and double-stranded DNA. After washing, the label incorporated at the damaged sites of the DNA is visualized by fluorescence microscopy. DAPI stain was added on top of TUNEL-treated cells.

### 2.11. Statistical Analysis

All experiments were repeated at least three times (unless indicated otherwise). All data were expressed as a mean ± standard error (SE) and the statistical differences between groups were evaluated using Student’s *t*-test for comparison between two groups or ANOVA test (or their non-parametric counterparts) for comparison between multiple groups. A *p* < 0.05 was considered statistically significant and SPSS software was used for the calculation of differences.

## 3. Results

### 3.1. Isolation and Identification of the Active Compound Structure

Bioassay-guided fractionation approach led to the isolation of the desired active compounds. HPLC injection of the crude extract revealed a chromatogram with three main peaks with retention times (RT) 15.6, 18.2, and 22.5 min (Appendix A). Purification of the extract by a series of chromatographic fractionations using RP-18 preparative column obtained several fractions. Each fraction that exhibited promising activity in vitro was further separated using various available chromatographic techniques. All isolated fractions were examined in vitro using an XTT assay for evaluation of biological activity. 

The purified compound with the ultimate anti-cancer activity appeared at RT of 15.8 min in the HPLC chromatogram with more than 90% purity (Appendix A). LC–MS of the compound shows two peaks. The main peak at the mass spectra appeared at *m*/*z* = 447.2382 (M + H^+^) with the calculated molecular formula of C_25_H_34_O_7_, and the second peak appeared at *m*/*z* = 429.2275 (M + H^+^-H_2_O) with the calculated molecular formula of C_25_H_32_O_6_ (elimination of water, H_2_O) (Figure 1). By using one- and two-dimensional NMR analyses (^1^H-NMR, ^13^C-NMR, ^13^C-DEPT135, COSY, HSQC, and HMBC, Appendix A), the structure of the active molecule (peak 15.8 min in HPLC) was elucidated as striatal C (Figure 2 and Table 1). ^13^C-NMR spectra of the active compound show twenty-five carbons; two of them are carbonyls (C14 and C3′), four carbons with double bonds (C3, C4, C11, and C12), and one acetal carbon (C1′) (Table 1). ^13^C-DEPT135 analysis showed the presence of four methyl groups (CH3), five methylene (CH2) groups, nine methine groups (CH), and seven quaternary groups (Table 1). According to the LC–MS and NMR spectroscopic data, the active compound (peak 15.8 min in HPLC) was identified as striatal C (Table 1, and Figure 2). The peak at 18.2 min was elucidated as striatal D (data not shown).

### 3.2. The Effect of the Isolated Active Compound on Cell Viability

The effect of the purified fraction, RT16, which contains striatal C was examined on viability of the human pancreatic cancer cell lines HPAF-II (Figure 3A) and PL45 (Figure 3B) in vitro. The cells were treated with different doses, 2.5–10 µg/mL, of RT16 and crude extract for 8, 12, and 24 h. The results indicated that RT16 significantly reduced cell viability of HPAF-II and PL45 in a dose-dependent manner, as well as the crude extract. Interestingly, the maximal effect was reached after 8h of treatment. Exposure of the cells to 10 µg/mL RT16 fraction for 8 h inhibited growth by 85% and 75% of HPAF-II and PL45 cells, respectively (*p* < 0.001). Whereas treatment of the cells with the *CS* extract (10 µg/mL) decreased viability by ~60% (*p* < 0.001).

In order to exclude the possibility for cytotoxic effects of RT16 on the cells, LDH leakage into the medium was measured following treatment with different concentrations of RT16 for 24 h. The results indicated that the effective doses of RT16, 2.5, 5 and 10 µg/mL, and 200 µM etoposide, a chemotherapy medication, have no significant effect on the release of LDH from the cells into the media. However, all treatments were found significantly different from the positive control (treatment of the cells with 2% Triton X-100) as detected by two-way ANOVA (*p* < 0.001) (Figure 4). These results may indicate that the effective doses of RT16 are not toxic to the cells.

### 3.3. The Effect of RT16 Fraction on Cell Cycle Progression

Treatment of the cells with RT16 resulted in significant reduction in cell viability. Therefore, it was needed to test the effect of RT16 on cell cycle progression. Cells were treated with different doses of RT16, 2.5, 5 and 10 µg/mL, or with 200 µM of etoposide for 4 h. Results indicated that exposure of the cells to RT16 significantly resulted in accumulation of the cells in the sub-G1 phase of the cell cycle (Figure 5). These results indicated that RT16 may induce apoptosis.

### 3.4. The Effect of the Isolated Active Compound on Induction of Apoptosis

In order to confirm that the accumulation of cells in the sub-G1 phase of the cell cycle is a result of induction of apoptosis, PL45 and HPAF-II cells were stained with FITC-labeled annexin-V and PI and analyzed by flow cytometry, as described under “Section 2”. For both cell lines, treatment for 4 h yielded an increase in apoptotic cells. Exposure of PL45 and HPAF-II cells to 10 µg/mL of purified RT16 fraction resulted in 52% and 32% apoptotic cells, respectively. Exposure of the cells to etoposide (200 µM) resulted in 6% and 18.5% apoptotic cells in HPAF-II and PL45, respectively, (HPAF-II: *p* = 0.32 and PL45: *p* < 0.001) (Figure 6).

For additional characterization of apoptotic cells, TUNEL assay was conducted, as described under “Section 2”. HPAF-II and PL45 cells were treated with purified RT16 fraction, *CS* extract (10 µg/mL), or etoposide (200 µM) for 4 h. Cells were labeled with TUNEL and DAPI and apoptotic cells were observed under fluorescent microscope. Cells treated with 10 µg/mL of the RT16 fraction presented typical apoptotic morphology, such as shrinkage of the cell and condensation of nuclear chromatin, as well as they exhibited the biochemical hallmark of apoptosis;internucleosomal DNA fragmentation in a more distinct manner than the *CS* extract- or the etoposide-treated cells (Figure 7).

## 4. Discussion

Previously, we found that ethyl acetate culture liquid extract from *Cyathus striatus* (*CS*) inhibited cell growth and induced apoptosis of human pancreatic cancer cells in vitro and inhibited tumor growth in vivo [22,23]. Based on these results, we aimed to identify the structure of the active compound on the *CS* extract. Fractionation studies and chemical analysis indicated that fraction RT16 contains one compound that was characterized as striatal C with the formula of C_25_H_34_O_7_. Moreover, we found that the purified RT16 fraction inhibited cell growth and induced apoptosis of cancer cells originated from human pancreatic cancer. In addition, the results indicated that effective doses of the RT16 fraction are not toxic to the cells, as well as the *CS* extract. Previously, we found that the *CS* extract effect is mediated by the induction of apoptosis via activation of both the intrinsic and extrinsic caspase pathways. Moreover, RNA-seq trascriptome analysis indicated that the *CS* extract may induce apoptosis via endoplasmic reticulum stress through the PERK/eIF2α/ATF4/CHOP pathway [23].

*Cyathus striatus* has proven to be a rich source of bioactive chemical compounds, as well as other medicinal mushrooms, that helps modulate the immune system, possess anti-cancer activity, and anti-oxidant properties [6]. It was reported that *CS* produces indolic substances, as well as a complex of diterpenoid antibiotic compounds, collectively known as cyathins [20,27]. Other study revealed the indole substances, known as striatins (A, B, and C), have antibiotic activity against *Fungi Imperfecti* and various Gram-positive and Gram-negative bacteria [21]. In addition, *Cyatus striatus* also produces sesquiterpene compounds called schizandronols [28]. It also contains the triterpene compounds glochidone, glochidonol, glochidiol and glochidiol diacetate, cyathic acid, striatic acid, cyathadonic acid, and epistriatic acid [29].

Epidemiological studies indicated that there is a correlation between cruciferous vegetable intake, vegetables that are rich in indole derivatives, and the development of cancer. Therefore, natural indole derivatives became very important compounds in drug-discovery development. Indoles represent a very important class of molecules that play a major role in cell biology and cancer research [30]. Previously, we found that indole derivatives—indole-3-carbinol (IC3) and 3-3′-diindolylmethan (DIM)—inhibited cell proliferation of human breast and prostate cancer cells in vitro [31,32]. Moreover, in vivo studies indicated that indole derivatives are not toxic, inhibit tumor growth, and have preventive effects against the development of prostate cancer in animal models [33,34].

Other studies indicated that indole derivatives represent anti-proliferative activities against hepatic and breast cancer cells. By studying the mechanism of action, it was reported that indole derivatives inhibit tubulin polymerization which leads to mitotic spindle formation disruption, cell cycle arrest in the G2/M phase, and induction of apoptosis [35]. Moreover, it was indicated that I3C and DIM modulate expression of a number of miRNAs. The number of target genes is important in the regulation of the cell cycle and apoptosis [36].

Triterpenes have numerous biological activities such as anti-cancer, anti-inflammatory, anti-oxidative, anti-viral, anti-bacterial, and anti-fungal properties [37]. Moreover, it was found that edible mushrooms play a role in preventing neurodegenerative diseases, such as Parkinson’s and Alzheimer’s. The mechanisms of action include reducing oxidative stress, neuroinflammation, and modulation of acetylcholinesterase activity [6]. Another study examined four mushrooms (*Fomitopsis pinicola*, *Hericium erinaceus*, *Inonotus obliquus*, and *Trametes versicolor*), commonly used in Asian and Far Eastern folk medicine, which targeted several biological processes relevant to cancer treatment [7]. In addition, it was reported that production of the nutritionally and medically rich oyster mushroom *Pleurotus Florida* induces apoptosis in human colon cancer cells [8].

In the present study, we identified striatal C in *CS* extract that exhibited anti-cancer properties. Striatal C induced apoptosis in human pancreatic cancer cells; therefore, striatal C can be added to the list of anti-cancer compounds in *CS*.

Pancreatic cancer is a disease with very poor prognosis and remains one of the deadliest cancer types world-wide [38]. According to the Global Cancer Observatory (GLOBOCAN), an estimated of 495,773 patients were newly diagnosed with pancreatic cancer worldwide in 2020 [38]. Therefore, as a malignant tumor with poor prognosis, improving the overall survival rate of pancreatic cancer patients is still a major challenge. Most patients lack sensitivity to most standard therapies, therefore, there is a need to develop new therapeutic strategies.

The anti-tumor activities of striatal C that was purified from the *CS* extract, as well as the proven utility of its anti-mitotic compounds against cancer cells, make striatal C an interesting compound with potential chemotherapeutic effects that may merit further research.

## 5. Conclusions

Ethyl acetate extracts from *Cyathus striatus* media inhibited cell growth of human pancreatic cancer cells and induced apoptosis through extrinsic and intrinsic pathways. Moreover, the *CS* extract inhibited tumor growth of pancreatic cells in animal models. The present study identified the chemical structure of striatal C, the main active compound in the *CS* extract. The results indicated for the first time that striatal C induced apoptosis in human pancreatic cancer cells. Further studies should be designed in order to clarify striatal C′s molecular mechanism of action. In addition, further studies are needed in order to develop a method for the synthesis of striatal C in order to use it in clinical studies for the treatment of cancer.

## Figures and Tables

**Figure 1 molecules-27-02746-f001:**
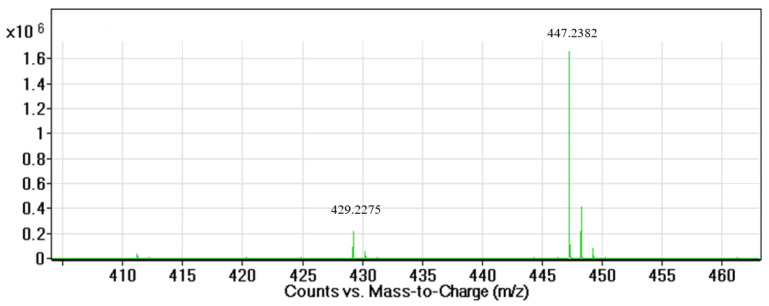
Mass spectra of the purified compound using QTOF LC–MS with ESI+ scan mode.

**Figure 2 molecules-27-02746-f002:**
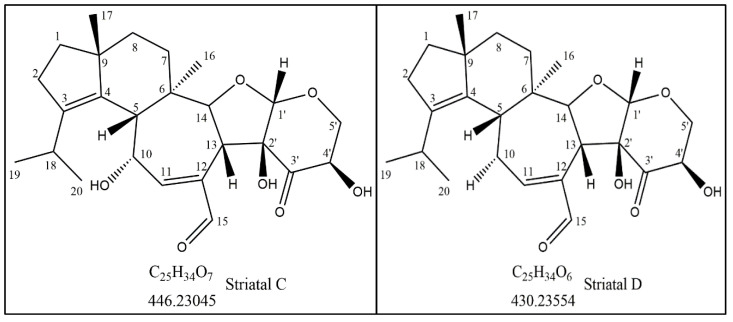
Structure of striatal C at 15.8 min and striatal D at 18.2 min in the HPLC chromatogram.

**Figure 3 molecules-27-02746-f003:**
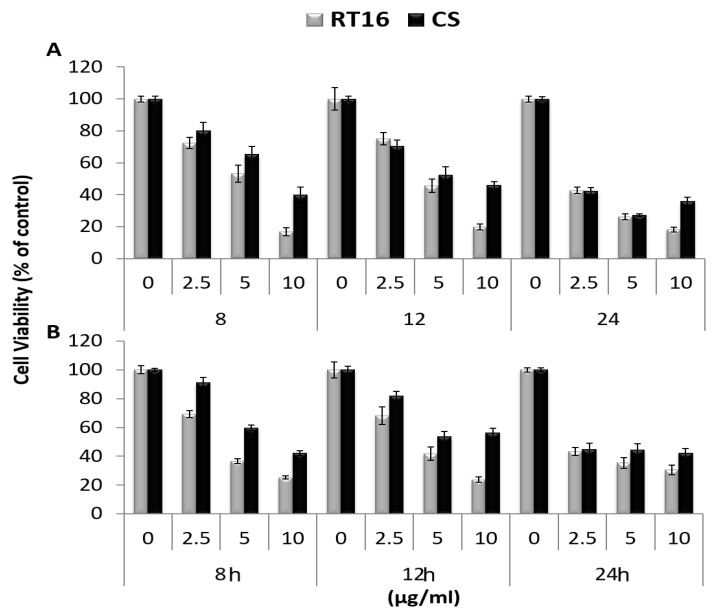
The effect of RT16 and *CS* extract on cell viability of HPAF-II (**A**) and PL45 (**B**) cells originated from human pancreatic cancer. Cells were seeded in a 96-well plate (10^4^ cells/mL) for 24 h, then treated with RT16 or *CS* extract (2.5–10 µg/mL), and incubated for 8, 12, or 24 h. Cell viability was measured using an XTT assay. The data presented are average of at least three independent experiments—3–5 repeats each (means ± SEM) and are expressed as percentage of the respective vehicle-treated control. Statistical significance was determined by two-way ANOVA (*p* < 0.001).

**Figure 4 molecules-27-02746-f004:**
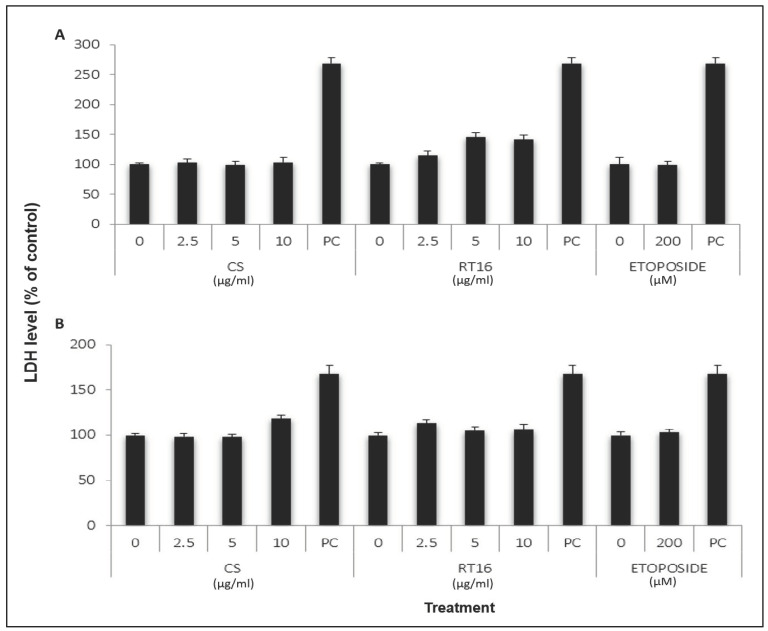
The effect of RT16, *CS* extract, and etoposide on LDH levels of HPAF-II (**A**) and PL45 (**B**) cells. The cells were seeded into a 96-well plate (10^4^ cells/mL) for 24 h and then were treated with various concentrations of RT16, *CS* extract, or 200 µM etoposide and incubated for 24 h. At the end of the treatment, LDH leakage was measured, as described under “Material and Methods”. Results presented an average of three independent experiments, three repeats each (mean ± SEM), and expressed as percentages of the control (non-treated cells). All treatments were found significantly different than the positive control (Triton X-100) which was detected by two-way ANOVA (*p* < 0.001).

**Figure 5 molecules-27-02746-f005:**
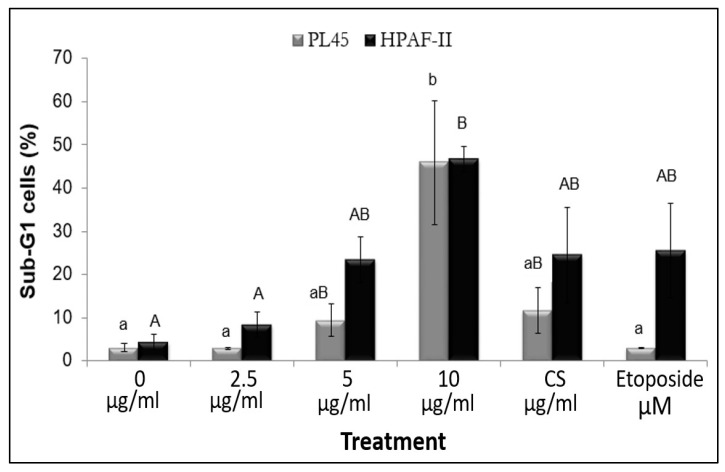
Analysis of cell cycle distribution. HPAF-II and PL45 cells were exposed to medium (control), RT16 (2.5, 5, 10 µg/mL), *CS* extract (10 µg/mL), or etoposide (200 µM) for 4 h. Cells were then harvested, fixed, and stained with PI and subjected to cell cycle analysis using FACS Calibur. The distribution and percentage of cells in the sub-G1 phase of the cell cycle are presented. Results presented are an average of at least three independent experiments, 3–5 repeats each (means ± SEM). Statistical significance was determined by one-way ANOVA followed by post hoc analysis, *p* < 0.001. Each letter represents relations to a fellow cell line in the same treatment period.

**Figure 6 molecules-27-02746-f006:**
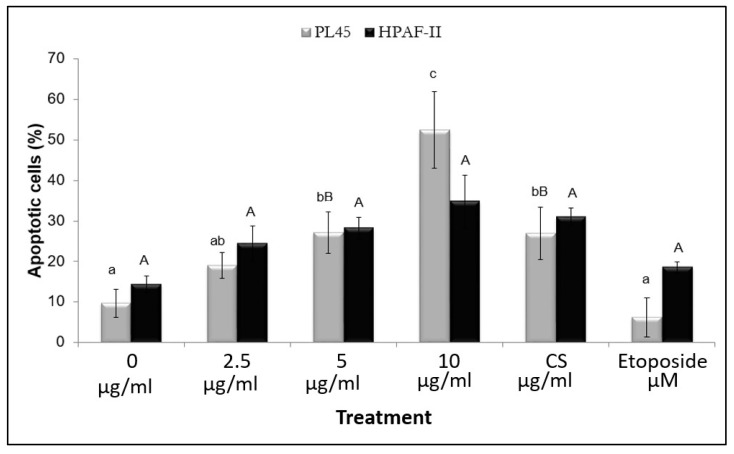
Annexin-V/PI staining for apoptosis. HPAF-II and PL45 cells were exposed to either medium (control), RT16 (2.5, 5, 10 µg/mL), CS extract (10 µg/mL), or etoposide (200 µM) for 4 h, and then stained with Annexin V-FITC and propidium iodide. A total of 5 × 10^5^ cells/mL were counted and collected for flow cytometry analysis. All results represent means ± SEM of three independent experiments. Statistical significance was determined by one-way ANOVA followed by post hoc analysis, *p* < 0.001. Each letter represents relations to fellow concentrations in the same cell line.

**Figure 7 molecules-27-02746-f007:**
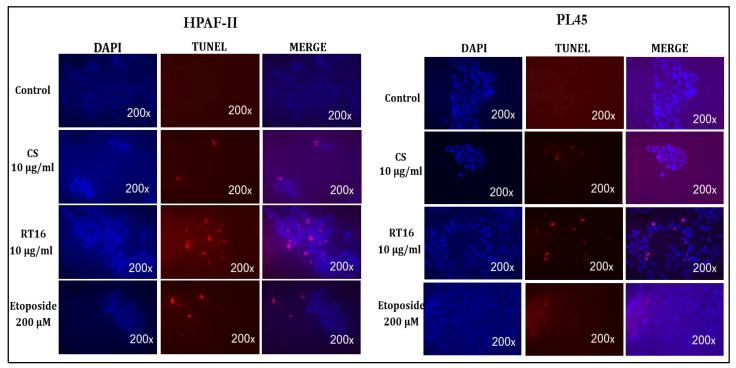
TUNEL and DAPI staining for apoptosis detection in HPAF-II and PL45 cells. A total of 3 × 10^4^ cells were seeded on chamber slides (Nunc, Denmark). After 24 h, the cells were treated with RT16 or *CS* extract for 4 h at a concentration of 10 µg/mL. At the end of the treatment, the cells were stained with TUNEL (Roche Applied Science, Mannheim, Germany) and DAPI (Vector laboratories Inc., Burlingame, CA, USA) and analyzed under fluorescent microcopy (×200 magnification).

**Table 1 molecules-27-02746-t001:** Assignment of the signals in the one- and two-dimensional NMR spectra of the isolated active compound (striatal C). The ^1^H- and ^13^C-NMR spectra were measured at 600 MHz and coupling constants are in Hz.

Carbon Number	^13^C-NMR (ppm)	^a^ HSQC ^1^H NMR Shifts (ppm) and Multiplicity	^b^ Number of Protons	^13^C-NMR DEPT135	Adjacent ^1^H-COSY (ppm)	HMBC Correlations
C1	39.22	1.72t	2	CH2	2.3	
C2	28.85	2.31t	2	CH2	1.72	C3, C4
C3	140.16	-	-	-	-	
C4	134.94	-	-	-	-	
C5	46.02	2.17	1	CH	-	C3, C4, C6, C7, C11, C16
C6	42.15	-	-	-	-	
C7	28.78	1.58	2	CH2		C6, C9
C8	36.60	1.57	2	CH2		C4, C6, C9
C9	49.17	-	-	-	-	
C10	68.77	4.96d	1	CH	6.90	C4, C5, C6, C11, C12,
C11	154.64	6.90dd	1	CH	4.96	C15, C5
C12	143.12	-	-	-	-	
C13	45.88	3.27, 3.28d	1	CH	4.5	C6, C11, C12, C14, C2′, C3′
C14	86.57	4.52, 4.54d	1	CH	3.27	C5, C7, C12, C16
C15	197.01	9.3s	1	CH	-	C12, C13
C16	21.11	1.22s	3	CH3	-	C5, C6, C7, C14
C17	23.88	0.96s	3	CH3	-	C1, C4, C8, C9,
C18	26.35	3.0q	1	CH	0.98, 1.04	C3, C2, C19, C20
C19	21.87	0.98d	3	CH3	3.0	C3, C18, C20
C20	21.62	1. 04d	3	CH3	3.0	C3, C18, C19
C1′	108.02	5.29s	1	CH	-	C14, C5′, C3′
C2′	83.50	-	0	-	-	
C3′	204.30	-	0	-	-	
C4′	75.00	4.03	1	CH	3.7	
C5′	68.62	4.30, 4.28, 3.80, 3.78dd	2	CH2	4.03	C4′, C3′

^a^: This column represents ^1^H NMR chemical shifts and multiplicity; the ^1^H NMR shifts correlated with the ^13^C NMR chemical shifts using HSQC 2D-NMR spectral analysis. ^b^: this column represents the number of protons obtained from peak integration from the ^1^H NMR spectra.

## Data Availability

The data presented in this study are available on request from the corresponding author.

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
