# Peer review of "Striatal Isolated from Cyathus striatus Extracts Induces Apoptosis in Human Pancreatic Cancer Cells"

_molecules, 2022, doi:10.3390/molecules27092746_

Round 1

Reviewer 1 Report

The authors successfully isolated the major compounds in the ethyl acetate fraction of Cyathus Striatus via biologically guided fractionation and were identified as striatal C and D using NMR. Please find below comments for your consideration  

Critique

  • I would love to see the different NMR spectra in the supplementary to increase the identification confidence of the readers.
  • Why are the biological activities of the pure compounds not indicated? This critical point is to be added to evaluate the validity of the fractionation workflow
  • Poor presentation of the results (Figures/tables)

Other inquires

  • Why was 1% formic acid mobile phase B used for the HPLC analysis, not 0.1% like mobile phase A?
  • Legends of the tables (P6) should be above the table unless otherwise indicated by the journal.
  • Please indicate Table 1 data refer to Strital C or D?
  • Please amend HSQC reported data to list the chemical shift and multiplicity which should be reflected in the headings! Furthermore, HMBC in the same table is not a chemical shift. It indicates the correlations and should not be expressed in ppm. A proper table header and explanatory footnotes may be needed.
  • Figure 3 is missing the statistical significance derived from ANOVA testing. Also, please amend the x-axis title to concentration ug/ml as it is not RT16 only.
  • Figure 4 is missing the statistical significance derived from ANOVA testing and missing A and B labels and X-axis is to be labelled concentration and put the units next to the treatments. Please spell out PC in the legends.
  • The results of the LDH in P7 (L222 -228) are to be revisited. It is not in agreement with the top panel of Figure 4, where the response to 5 and 10 ug/ml RT16 may be statistically significant compared to negative control and CS counterparts. Unfortunately, all stats are missing.
  • Add the RT16 next to the concentrations in figure 5 and figure 6 and spell out any contracted names such as “Etop” in the legends.

Minor

  • Line 36 p1; remove the spelled out higher Basi. Mushrooms as it was spelled out the line before. Same for uncontracted low-molecular-weight (LMW) at line 57.
  • Line 55; please amend “fungi imperfect” to either imperfect fungi or fungi imperfecti
  • Line 80; spell out DDW at its first appearance >”double distilled water”
  • Line 97; plz amend 300oC to 300°C
  • Line 180; the time unit is missing “15.8 min”
  • Line 184; remove the extra bracket in “((elimination of water, H2O)”
  • Line 201; remove the molecular formula after the figure 2 legend

Author Response

We thank the reviewer, all comments were checked and corrected in the manuscript

Reviewer 2 Report

  1. Please elaborate the introduction section and cite some relevant and recent references in this section.
  2. Improve the clarity of fig.1. Some graphical components are not visible.
  3. Please give exact p values for the histogram.
  4. Please give 4x, 20x, 40x and 100x magnification for fig. 7. Scale bar should be included in all of these images.
  5. Elaborate the discussion section by citing the most recent articles related to your manuscript.
  6. Cite PMID: 34412526, PMID: 33601145, PMID: 33196993, and PMID: 33083576 in your manuscript.
  7. The manuscript should be checked by an authentic plagiarism checker before publication.

Author Response

We thank the reviewer for his constructive comments

Round 2

Reviewer 1 Report

Happy with the amendments where a final proofreading may be needed.

This manuscript is a resubmission of an earlier submission. The following is a list of the peer review reports and author responses from that submission.